# Nationwide monthly burned area monitoring in Indonesia using Sentinel-2

**David L. A. Gaveau**[1,2]*, **Adrià Descals**[3], **Mohammad A. Salim**[1]

**1** The TreeMap, Montpellier, France, **2** Jeffrey Sachs Center on Sustainable Development, Sunway University, Kuala Lumpur, Malaysia, **3** Department of Biology, University of Antwerp, Wilrijk, Belgium

* d.gaveau@thetreemap.com

## Abstract

Wildfires pose a major challenge for many nations. Rapid mapping of their extent is key to evaluating their impacts. We present the first operational monthly burned-area processing chain for Indonesia, based on largely automated processing of Sentinel-2 imagery in Google Earth Engine. Our approach uses a Random Forest applied to Sentinel-2 imagery and integrates FIRMS fire hotspots to reduce false positives. The resulting 20-m monthly burned-area maps cover the entire country. From January 2019 to December 2024, fires burned a cumulative 5.62 million hectares (Mha), including 2.92 Mha that burned once and 1.12 Mha that burned multiple times. This represents a total burned extent of 4.04 Mha. Compared to the MCD64A1 product, our dataset detects more burns with higher spatial detail and accuracy. In total, 122,164 hectares of primary humid forest burned, representing 2.2% of the burned area. In 2019 and 2023, fire activity accelerated around July and peaked in September–October, coinciding with Oceanic Niño Index (ONI) values ≥ +0.5 °C and Indian Ocean Dipole (IOD) values ≥ +1.5°C. In contrast, neutral or negative phases from 2020 to 2022 corresponded with minimal burning. The year 2024 recorded intermediate fire activity without strong climatic anomalies. These findings confirm that climatic anomalies are associated with fire activity in Indonesia, reaffirming the importance of ONI and IOD for early warning. Our results suggest that prevention efforts are limiting forest fires, as burns in 2019 and 2023 remained lower than during earlier events. Monthly burn-scar updates are available on Nusantara Atlas (www.nusantara-atlas.org), an open-access platform for monitoring deforestation in Southeast Asia.

## Introduction

Wildfires damage terrestrial ecosystems, threaten human lives, and disrupt economies [1]. They often spread beyond control, especially in fire-prone regions with degraded ecosystems [2,3]. This contrasts with traditional controlled burns, which have long been used by farmers to manage farmland. Rapid and accurate mapping

**Data availability statement:** The data underlying the results presented in the study are available from Zenodo at: https://zenodo.org/records/19104282.

**Funding:** The author(s) received no specific funding for this work.

**Competing interests:** The authors have declared that no competing interests exist.

of burned areas is necessary for quantifying the extent of the damage, understanding the causes, and guiding response strategies.

Indonesia is a major wildfire hotspot in the equatorial region [4]. Recurring forest fires have become an international concern because the emitted smoke regularly blanket Southeast Asia in toxic haze for weeks. Their frequency and intensity have sharply increased since the late 20th century, particularly in Kalimantan after 1980, due to large-scale land-use change [5,6]. While El Niño and the Indian Ocean Dipole create dry conditions that favour fire spread [7,8], humans activities intensified risk and triggered more fire events [5]. In particular, large-scale degradation and deforestation of the humid forests of Equatorial Asia have accentuated wildfire occurrence. This began with large-scale timber extraction in the 1970s followed by conversion to industrial plantations [9,10]. Logging roads, peatland drainage canals, Food Estates, and industrial plantations have further fragmented vast 'otherwise forest-resistant' humid forests, and lowered water tables in peat soils, making them highly flammable during droughts [11,12]. Even well-resourced oil palm and pulpwood estates, where fire risk declines once land conversion ends, regularly suffer significant losses of productive plantations to wildfire [13].

Land burning is illegal in Indonesia under Law No. 32 of 2009 on Environmental Protection and Management, with limited exceptions for traditional smallholder practices. Despite this ban, fire use remains widespread because enforcement is weak and land disputes persist [13]. Fire is employed by a range of actors, including small farmers, farmer associations, land speculators, and plantation companies as a low-cost method to clear land, dispose of woody debris, and fertilize soils [14,15]. In this context, detailed and rapid burned area assessments, when combined with contextual spatial data such as concession boundaries, cadastral maps, administrative boundaries, peatlands extent and drainage canals provide evidence for forensic investigations, help authorities enforce legislation and impose fines more effectively [13].

Satellite remote sensing provides consistent observations over large areas and, thus, has become an important tool for mapping burned areas [16]. Previous studies have used MODIS sensors for detecting burn scars [17]. The most used burned area products are the MCD64A1, FireCCI51, and C3SBA10 due to their global coverage [17–19]. However, these datasets miss small fires particularly in heterogeneous landscapes and smallholder agricultural areas [20]. Consequently, this limitation has driven the development of more detailed burned area maps using higher-resolution satellites (10–30 m) [21,22].

Research has shown that Sentinel-2 (10–20 m resolution) and Landsat (30 m) improve the detection of small burn scars in complex tropical landscapes. In Indonesia, Landsat has been extensively used by the Directorate of Forest and Land Fire Control [23], which is under the Directorate General for Forestry Law Enforcement of the Ministry of Forestry and is responsible for forest-fire control policy. Similarly, Sentinel-2 data are increasingly used to detect burned areas in parts of Indonesia [24–27] and to generate national-level burned-area datasets in sub-Saharan Africa [20].

Despite these advancements, mapping burned areas on monthly basis remains challenging in tropical regions. Persistent cloud cover and heterogeneous land cover types complicate the detection of burn scars and reduce the consistency and accuracy of the maps [28,29]. Moreover, tropical vegetation can regrow rapidly after burning and burn scars may become undetectable within a few weeks. This further complicates the accurate monitoring of fires, especially over areas where satellite observations are frequently obscured by clouds.

Our previous study addressed these challenges by developing a methodology based on a machine learning model (Random Forest) that classified Sentinel-2 data [30]. The workflow was implemented in Google Earth Engine [31] and produced national-scale annual burned area maps. The method represented a major advancement because it produced improved burned area maps that were able to detect small fires and outperformed the MCD64A1 product [30]. Despite the improvements, the method was primarily designed for retrospective annual analysis. The increasing demand for timely geospatial data requires approaches that can deliver accurate burn-scar maps in near-real time. This can be achieved primarily by using fine-scale imagery from Sentinel-2, which offers a higher revisit frequency than Landsat. This operational approach would offer finer detail than the MCD64A1 product, at 500-m resolution, which is the most accurate global monthly burned area product currently in operation [17].

Recent studies have proposed methods for monthly burned-area mapping in Indonesia, using Sentinel-2 and other remote sensing inputs [26,32]. However, none have yet delivered an operational monthly burned-area mapping system for Indonesia. To address this gap, we present the first automated monthly burned-area processing chain for Indonesia. Our method integrates high-frequency Sentinel-2 time series imagery (every 2–5 days) with FIRMS (Fire Information for Resource Management System) daily fire hotspots [33]. These daily hotspots are derived from MODIS (Moderate Resolution Imaging Spectroradiometer) sensors onboard NASA's Terra and Aqua satellites and the VIIRS (Visible Infrared Imaging Radiometer Suite) sensors onboard the Suomi NPP and NOAA-20 satellites [34]. These instruments detect daily thermal infrared anomalies associated with active combustion on the surface of the earth. In this study, the fire hotspot detections serve as spatial and temporal filters for identifying candidate burned areas detected using Sentinel-2 imagery. The workflow is implemented in Google Earth Engine [31], where Sentinel-2 imagery is processed to generate monthly burn-scar maps at 20-meter resolution, with a minimum mapping unit of 6.25 hectares — aligned with the official minimum burn-scar size adopted by Indonesian authorities [35].

## Methods

### Summary of methods

A burned area is identified by alteration of vegetation cover and structure along with deposits of char and ash. We mapped such areas using a change-detection approach, i.e., by comparing Sentinel-2 top-of-canopy reflectance before and after a burning event [36].

The monthly burned-area processing chain is composed of three consecutive steps. Step 1 involves the processing of Sentinel-2 images into monthly cloud-free pre- and post-fire composites, which reflect the Earth's surface before and after the first day of the reference month. In this study, the month being processed in the processing chain is called the *reference month*. Step 2 entails the classification of the Sentinel-2 composites into monthly burned area maps using a 'Random Forest' classifier. Step 3 includes a post-classification procedure consisting of four sub-steps: a) removal of small patches of pixels (<6.25 ha) that were detected as burned by the 'Random Forest' classification, aligning with the Indonesian official minimum mapping unit requirements, b) morphological operations to refine the visual appearance of the classification layer, c) removal of 'burned' false positives using FIRMS data, and d) identification of burned areas that were also detected in preceding months. Steps 1, 2, 3a,b are modifications of the annual burned-area processing chain published previously [30], whereas steps 3c,d are additional steps specific for the monthly burned-area processing chain. Fourth, we assessed our monthly burned-area map, against the monthly MCD64A1 burned-area product [17], and against our previously published reference annual dataset [30] to gauge the reliability and accuracy of the three burned-areas

products. Details of the operational execution, automation, scheduling, and integration of the processing chain with the Nusantara Atlas platform are described in S1 Text and illustrated in Figure 1 in S1 Text.

## Sentinel-2 pre- and post-fire Sentinel-2 monthly composites

Here, we describe our procedure to create pre- and post-fire composites. Prior to creating the composites, we removed non-valid pixels using the Sentinel-2 imagery quality flag (this flag provides information about clouds, cloud shadows, and other non-valid observations) produced by the ATCOR processing chain and included in the atmospherically-corrected surface reflectance multispectral images of the Sentinel-2 A and B satellites Surface Reflectance products (Level 2A product) [37].

The pre-fire composite is built using imagery from a three-month window preceding the reference month (i.e., the month being analysed by the processing chain). For example, if September is the reference month, the pre-fire composite includes all valid observations from June 1st to August 31st. This relatively long compositing window was chosen to ensure sufficient cloud-free observations while capturing a stable representation of the vegetation and land cover prior to fire events. In tropical regions like Indonesia, where cloud cover is frequent, this approach reduces the risk of missing data due to persistent atmospheric interference. Moreover, since vegetation typically undergoes minimal short-term changes before a fire, this extended temporal window does not compromise the characterization of pre-fire conditions but instead increases compositing robustness.

The post-fire composite, by contrast, includes only observations acquired during the reference month (e.g., September 1st to 30th), ensuring that it captures fire-related spectral changes as close as possible to the time of burning. For both pre- and post-fire composites, we apply a median aggregation of all valid pixel values within the compositing window to reduce noise from outliers and residual artifacts.

This approach differs from our previous annual-burned-area processing chain [30], which used a pixel-wise moving window to determine the compositing period based on the minimum Normalized Burn Ratio (NBR) value observed each year. As a result, annual composites tend to capture burn scars accumulated over the entire fire season, while the monthly composites are temporally constrained to the reference month and are therefore more suitable for near-real-time monitoring (Figs 1 and 2).

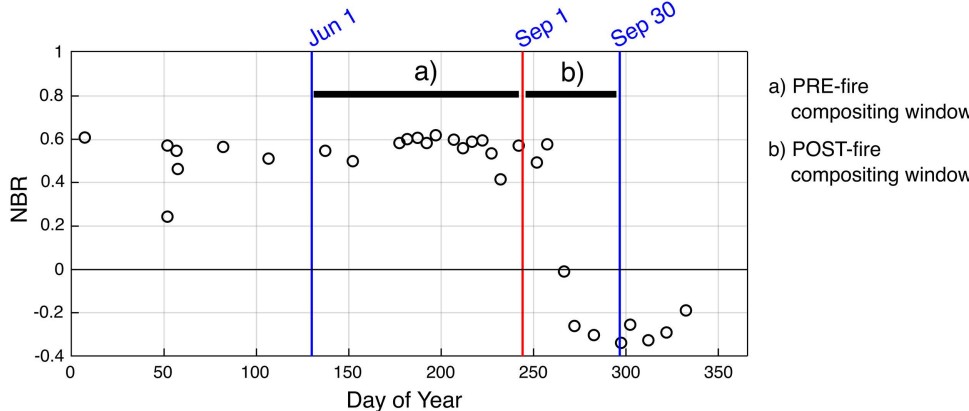

**Fig 1. Example of a Sentinel-2 Normalized Burn Ratio (NBR) time series.** The example corresponds to a single pixel (20 × 20 m) in Kalimantan, where the reference month being processed is September. The pre- and post-fire compositing windows used in the monthly burned-area processing chain are indicated.

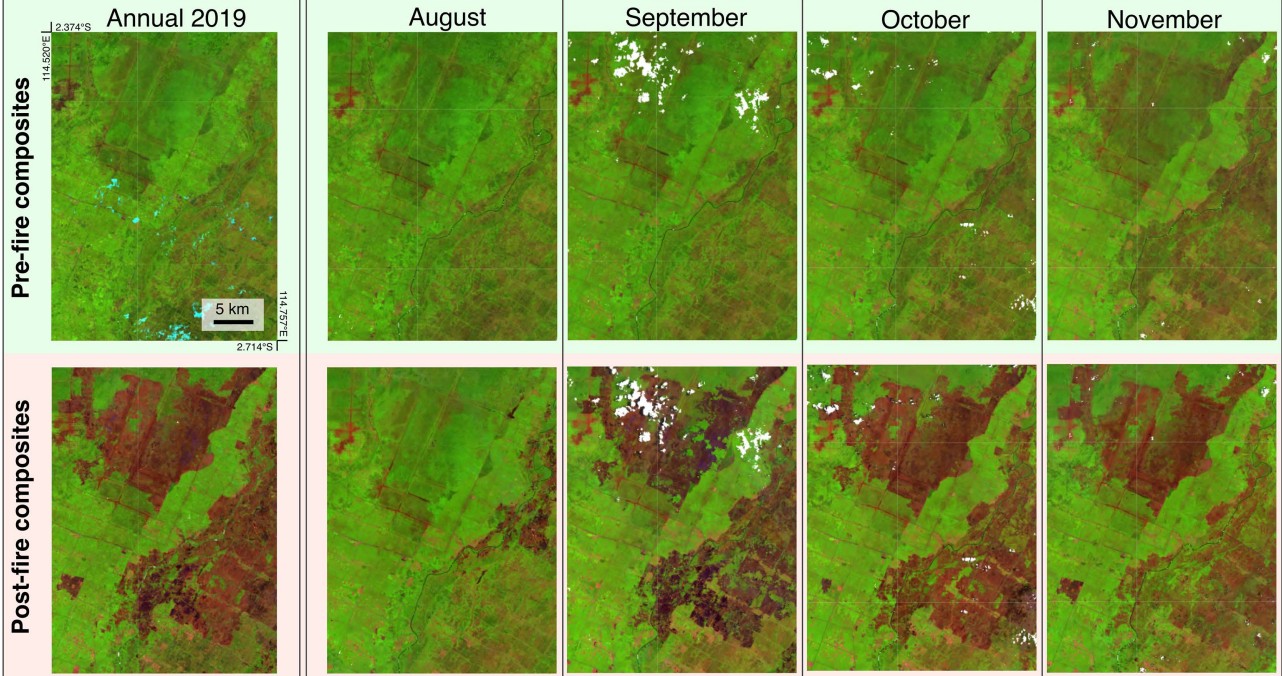

**Fig 2. Monthly versus annual Sentinel-2 pre- and post-fire composites over an area in Central Kalimantan.** Comparing Sentinel-2 pre- and post-fire monthly composites (August-November 2019) with the annual composites published previously [30]. Sentinel-2 composites are displayed in false colours (RGB: short-wave infrared, band 11; near infrared, band 8; blue: red, band 4). The images reveal large burn scars, visible as areas that have transitioned from green to dark brown/red tones. The Monthly sequence reveals the progression of the burn scars.

## Supervised burned/unburned classification

We used the 'Random Forest' [38], a supervised classification algorithm available in the Google Earth Engine, to determine whether the spectral changes observed by the monthly pre- and post-fire composites corresponded to a fire event, and subsequently classify burned areas. 'Random Forest' is a popular algorithm because it is a non-parametric model that can process large datasets and yield high accuracies without requiring a large amount of training data or rigorous hyper-parameter optimization. The ensemble nature of random forests also reduces the risk of overfitting.

The input variables used in the 'Random Forest' are the spectral bands of Sentinel-2 in the pre- and post-fire composites plus their respective NBR index. We excluded the bands at 60-meter spatial resolution (bands B1, B9, and B10) since these bands present a low spatial resolution for the aim of the study. Therefore, we used a total of 22 features: the NBR and bands B2, B3, B4, B5, B6, B7, B8, B8A, B11, and B12 and the NBR index of the pre- and post-composites. The 10-meter bands (B2, B3, B4, and B8) were resampled to 20 meters to align with the spatial resolution of B5, B6, B7, B8A, B11, and B12. The class predictions of the 'Random Forest' were 0 (class 'burned') and 1 (class 'unburned').

Supervised classification models require training data, that is, exemplary spectral signatures of 'burned' and 'unburned' lands in the present case, to guide the to reliably classify the target classes. In this study, we used the training dataset published previously [30], and collected additional training data to improve the classification of the monthly Sentinel-2 composites. The additional training points were collected to reduce false positives (classified as 'burned') over moist, vegetation-free soils, particularly in peatland areas during wet years (2020 and 2021) when mechanically cleared peatlands with exposed, water-saturated soils exhibit spectral signatures like those of burn scars. The final training dataset consists of 2,343 points; 852 points for the class 'burned' and 1,491 points for the class 'Unburned' (Fig 3).

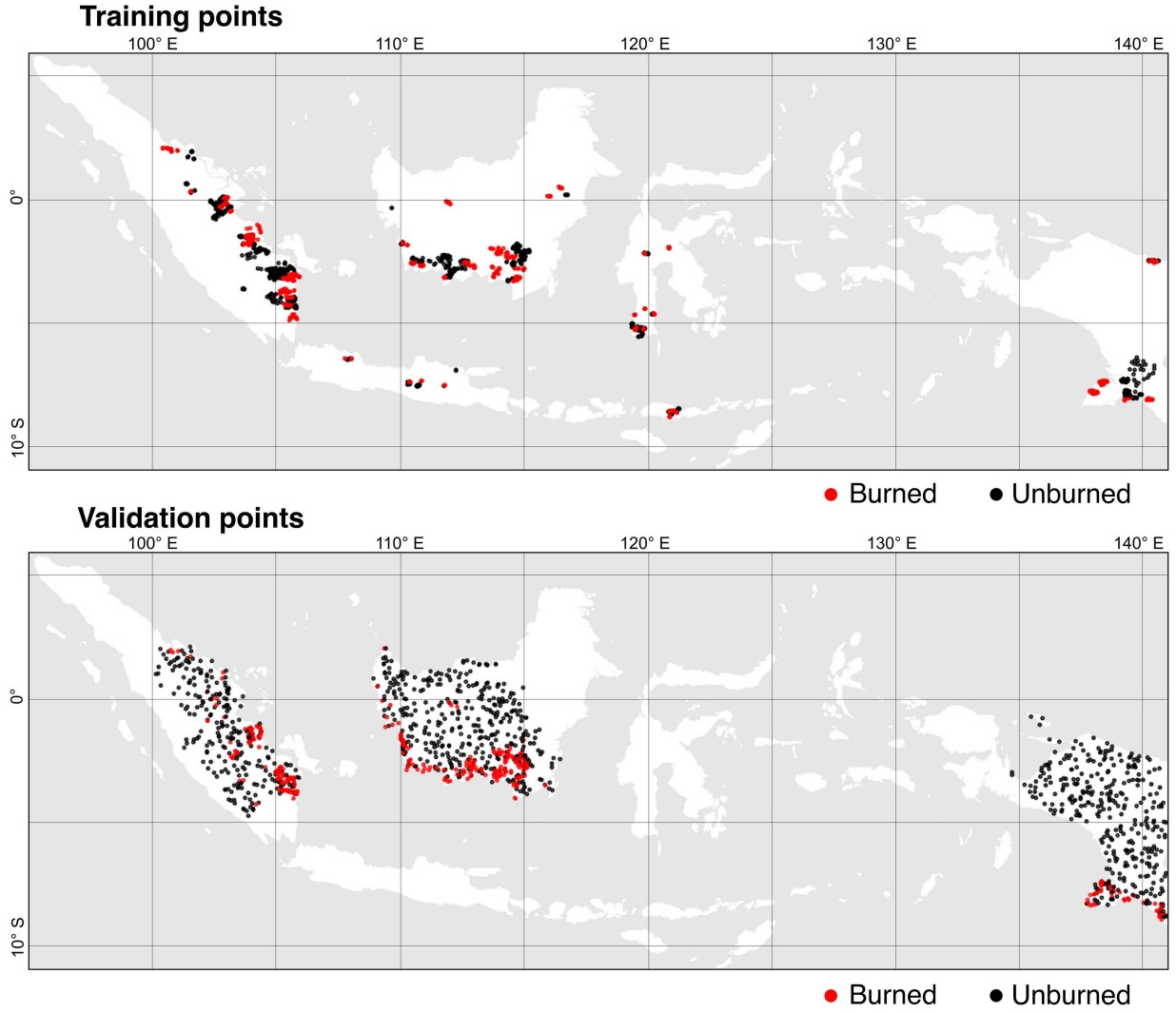

**Fig 3. Location of training and validation points labelled as 'Burned' and 'Unburned'.** The map above shows the 2,343 training points used to train the Random Forest. The map below shows the 1,042 reference points used to validate the burned area map.

## Post classification procedure

**Masking of small patches.** The first step in the post-classification involved removing small patches of pixels categorized as 'Burned.' The threshold was set to 6.25 ha, equivalent to 156 pixels in the 20-meter classification images. Small patches below 6.25 ha, including isolated pixels classified as 'Burned,' were removed to stay in line with Indonesia's ministry of forestry who adopted a minimum mapping unit for national burned area assessments of 6.25 ha.

**Morphological operations.** Some detected burned areas had unnaturally rough or blocky shapes that didn't match the true outlines of the fire scars. To correct this, morphological operations were applied to smooth and round the shapes of the burn patches and improving their visual realism. The process involved two dilation steps: i) opening using a 3x3 square kernel, which helps remove small noise and detach weak connections; and ii) closing using a 3x3 circular kernel, which fills small gaps and further smooths the shape. Additionally, a rule was applied to avoid misclassifying vegetated

areas as burned: If a pixel's post-fire NBR (Normalized Burn Ratio) was greater than −0.1, it was kept as 'unburned'. This threshold filtered out vegetated pixels, where burning is unlikely.

**Masking false positives with FIRMS daily fire hotspots.** Despite using an improved training dataset, false positives persisted, particularly during wet years, when peatlands with exposed, water-saturated soils exhibit spectral signatures like those of burn scars. To correct this, we used a post-classification procedure that filtered the false positives of class 'Burned' using daily fire hotspot data from FIRMS. FIRMS data comprises points where MODIS or VIIRS satellites detected thermal anomalies, indicating the presence active fires [33,39]. A hotspot point represents the centroid of a pixel in which a thermal anomaly was detected and, thus, the location of the fire is within the perimeter of the pixel. As a result, hotspot points have an uncertainty that is associated to the spatial resolution of the satellite sensors. The spatial resolution is approximately 1 km for MODIS and approximately 375 m for VIIRS. In the first step of this procedure, we buffered by 100 meters the pixels that we detected as burned in our monthly burned area classification. Then, we re-classified the 'Burned' pixels to 'Unburned' if no fire hotspot was present within the burn scar or in the buffering area. This rule ensured that only the patches of pixels where there is evidence of active fire were considered in the final burned area layer. The masking procedure uses daily FIRMS hotspot data taken in the three months before the reference month and the reference month. For instance, if September is the reference month, the hotspot points include FIRMS data detected from July 1st until September 30th.

**Masking repeated burning detection.** The burned area processing chain produces monthly burned area layers intended to capture fire activity within a specific reference month. However, because burn scars can remain visible in Sentinel-2 imagery for months, the same area may be repeatedly detected as 'Burned' across consecutive months, leading to potential double counting. To prevent this, we applied a filtering rule that reclassifies a pixel as 'Unburned' if it had already been detected as 'Burned' in any of the previous three months. For example, if a pixel was detected as 'Burned' in February 2019, and remained visible in March, April and May, it would not be counted again in those subsequent months (Fig 4).

## Burned-area map validation

To assess the spatial accuracy of our monthly burned-area product, we validated it against a randomly distributed reference dataset originally developed for the annual burned-area product published previously [30]. The validation follows best practices guidelines and employs a balanced sampling approach to minimize bias [40]. The reference dataset was created using stratified-random sampling of 'Burned' and 'Unburned' sites across seven fire-prone provinces in Indonesia. Each site (20 × 20 m) was visually interpreted using the original time-series Sentinel-2 imagery by three independent interpreters to detect fire activity based on the presence of flaming fronts, smoke, or charred vegetation.

In our previous publication, we sampled reference points to validate three burned area layers: our annual burned-area layer, the MCD64A1 annual burned-area product, and an annual burned-area map produced by the Indonesian Ministry of Environment and Forestry [30]. The validation dataset contained 1,168 reference points, including 280 'burned' and 888 'unburned' points. For this study, we adjusted the number of reference points to validate monthly burned-area layer. This correction was necessary to ensure a proper stratified random sampling, in which all pixels mapped with the same class have the same probability of being sampled. The correction consisted of i) randomly sampled 20 points within the area classified as 'Unburned' in the monthly burned-area layer and 'Burned' in the annual burned-area layer (area in light red in Fig 5); ii) randomly reduce the number of points classified as 'Burned' in the monthly burned-area layer and 'Unburned' in the annual burned-area layer to 4 points (area in cyan), and iii) randomly reduce the number of points classified as 'Burned' in the monthly and annual burned-area layer to 150 points (area in red). The final reference point dataset used for validating the monthly burned-area layer consisted of 1,042 points: 150 'Burned' points and 892 'Unburned' points (Fig 3, lower panel).

                                                            

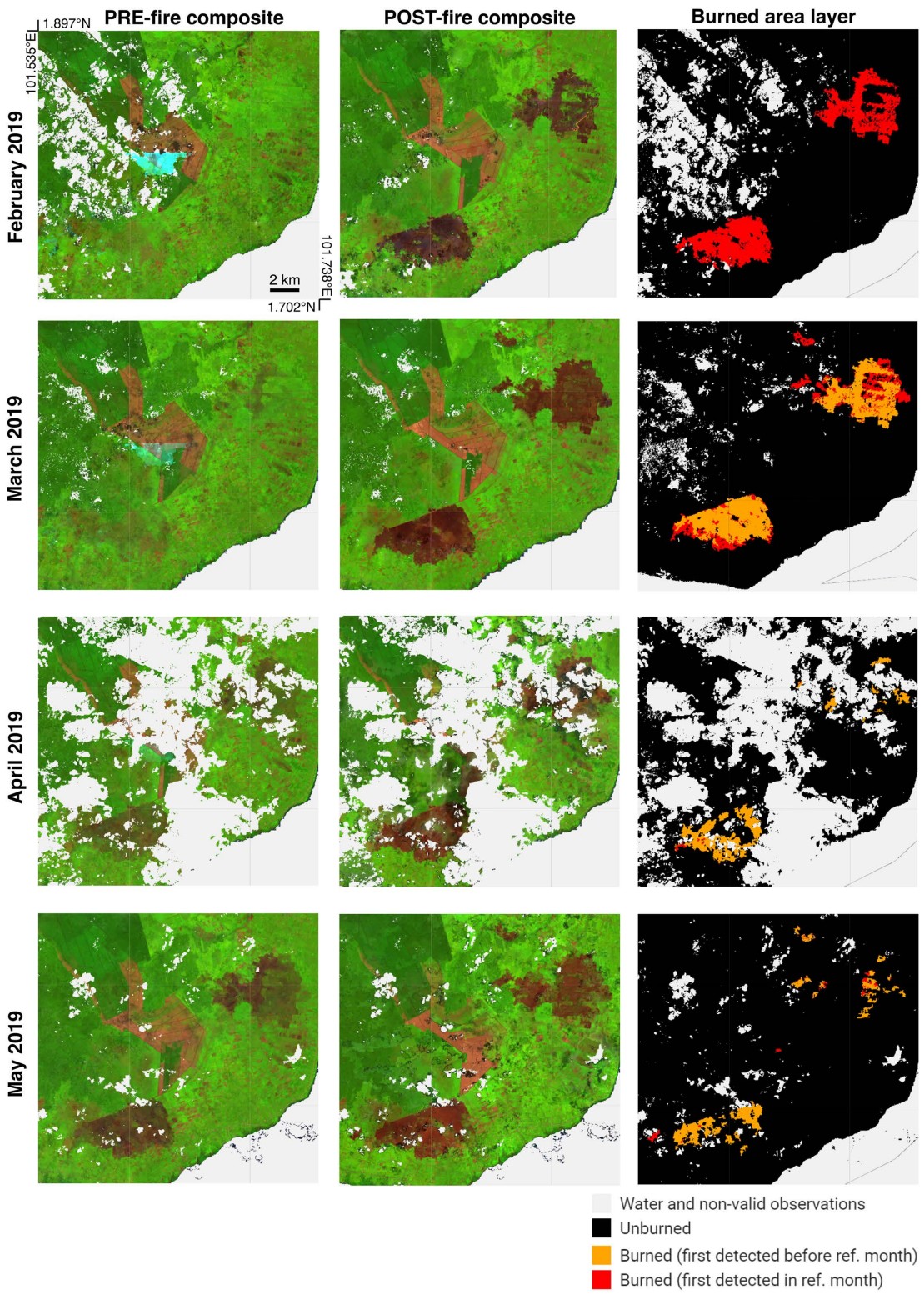

**Fig 4. Persistence of burn scar visibility leading to repeated monthly detection by Sentinel-2 imagery.** Pre- and post-fire Sentinel-2 monthly composites, and corresponding burned-area classifications for February, March, April, and May 2019 are shown over an area (*Pulau Rupat*) in Riau province, Sumatra where peatland burn scars remained visible several months after fire. Pixels that were detected as 'Burned' for the first time are shown in red, while pixels that were repeatedly detected as 'Burned' are shown in orange.

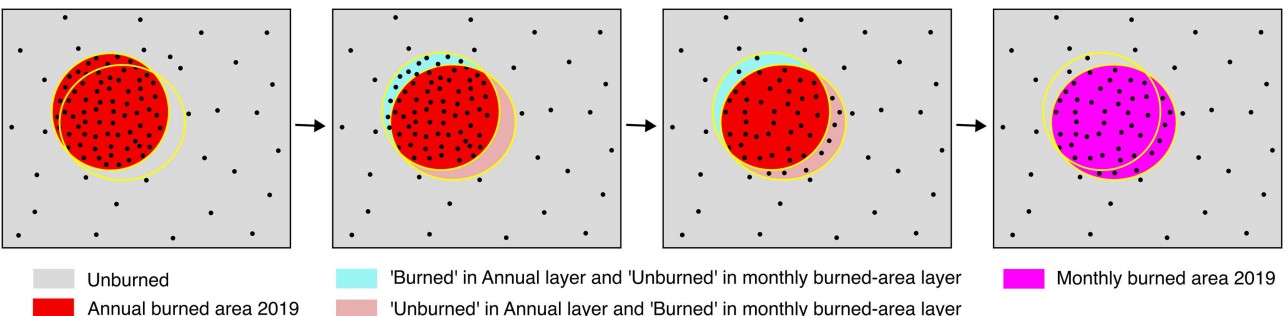

**Fig 5. Illustrative representation of validation points used for the monthly burned-area layer.** From the original 1,168 points, we adjusted the sample to ensure proper stratified random sampling in the monthly burned-area layer in 2019.

We computed overall accuracy (OA), producer's accuracy (PA), and user's accuracy (UA) with 95% confidence interval. Validation was conducted on the cumulative monthly burned-area map for 2019, which is the same year as the reference dataset. The validation metrics were also calculated for the annual burned-area layer and the MCD64A1 product.

To evaluate the temporal accuracy of the monthly burned-area product, we compared the month of detection against the independent reference dataset derived from visual interpretation of Sentinel-2 time series. This reference dataset consists of burn dates for 259 points across seven fire-prone provinces in Indonesia. These 259 points include all reference points classified as 'burned' in the burned area layer and with a true class of 'burned'. We did not apply the point correction used in the spatial validation illustrated in Fig 5 as a stratified sampling is not required for the temporal validation. For each site, we recorded the first month in which the burn scar was visible in the Sentinel-2 imagery. We then constructed a confusion matrix comparing the month assigned by the burned-area processing chain with the month determined by visual inspection. This comparison allowed us to assess how accurately the processing chain captures the timing of burn events at monthly resolution and to identify potential lags in the detection of the burn events.

We also evaluated burn size distributions across burned-area datasets to examine potential biases in fire detection. We compared the frequency distributions of burn scars among our Sentinel-2-derived burned-area maps, and the MCD64A1 dataset. Differences in burn size distributions were tested using the Kruskal-Wallis H test, followed by Kolmogorov-Smirnov and Mann-Whitney U tests to assess whether specific datasets over- or under-represented certain burn size classes.

## Results

### Burned-area detection

In six years, between 2019 and 2024, fires burned a cumulative 5.62 million hectares (Mha), including 2.92 Mha burned once and 1.12 Mha burned multiple times (Fig 6). This represents a total burned extent of 4.04 Mha.

The spatial distribution of burned area from 2019 to 2024 reveals a highly uneven pattern across Indonesia's regions and provinces (Fig 7-8). At the regional scale, Kalimantan, Nusa Tenggara, and Sumatra experienced the largest extent of burning, with each surpassing 1 million hectares (Mha). These three regions consistently recorded substantial fire activity across multiple years, particularly during peak fire seasons in 2019 and 2023. At the provincial level, Nusa Tenggara Timur, Kalimantan Tengah, Papua Selatan, and Sumatera Selatan stand out as the most affected, collectively accounting for 52.2% of total burned area during the 2019–2024 period.

When examined by year, national cumulative burn trends from 2019 to 2024 show notable interannual variability: peak burning occurred in 2019 and 2023, moderate levels were observed in 2020 and 2024, and fire activity was lowest in 2021 and 2022 (Fig 9). Our method detects significantly more burns than the MCD64A1 across all years. For example, in

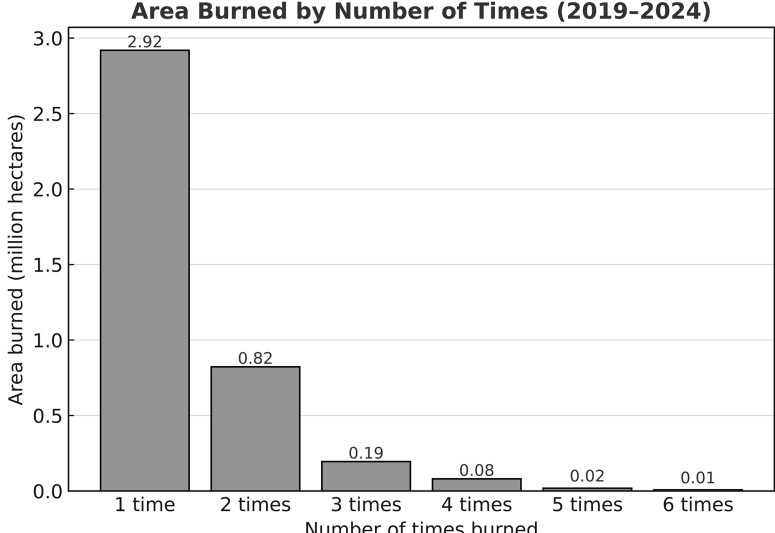

**Fig 6. Area burned by number of times between 2019 and 2024.**

2019, the burned area reached 2.64 Mha in our dataset compared to 2.09 Mha in MCD64A1; in 2023, the difference was more pronounced, with 1.70 Mha recorded in our study versus 1.11 Mha in MCD64A1. Even during low-fire years such as 2021–2022, MCD64A1 continued to underestimate burned area, reporting 0.05–0.07 Mha, compared to 0.06–0.11 Mha in this study. Fire activity in 2020 and 2024 was of intermediate intensity: in 2024, we recorded 0.49 Mha versus only 0.15 Mha in MCD64A1, and in 2020, the difference between both datasets was largest, with 0.62 Mha detected by our method compared to 0.18 Mha in MCD64A1.

Using a forest area mask representing conditions in January 2019, which we developed previously [41], we detected 122,164 ha of burned area in primary forests between 2019 and 2024, corresponding to 2.2% of the total burned area (5.62 Mha). MODIS reported a higher burned forest area (180,928 ha), which represents 5.0% of its total burned area. Regarding peatlands, the MCD64A1 product also reported a higher proportion of burning on peat (27.4%), whereas our estimate showed lower, but still substantial share of 16.1%. These results indicate that MCD64A1 consistently underestimates burned area across all years, which stems from its coarse spatial resolution and limitations in detecting small fires, leading to an incomplete representation of fire extent in Indonesia, and suggest that our maps capture a broader extent of burning, over both peatland and mineral soils, indicating improved detection across diverse land types.

### Fire activity and climatic anomalies

Analysis of monthly burned area across Indonesia from January 2019 to December 2024 reveals a strong association between fire activity and large-scale climatic anomalies (Fig 10). Two major fire seasons occurred in late 2019 and 2023, with monthly burned areas peaking in September and October. These peaks in fire activity closely align with positive phases of both the Indian Ocean Dipole (IOD) and the El Niño Southern Oscillation (ENSO), the latter indicated by positive values of the Oceanic Niño Index (ONI), which reflect El Niño conditions. In 2019, a strong positive IOD (+1.8°C) and a moderate El Niño (ONI reaching +0.5°C) coincided with the highest burned area recorded during the study period. Similarly, in 2023, a strong El Niño (ONI > 1.5°C from September to December) alongside a positive IOD (+1.5 °C), preceding the second largest fire season observed. In contrast, the period from 2020 to 2022 was characterized by neutral to negative IOD and ONI phases, coinciding with minimal fire activity, with peak monthly burned areas consistently below 0.2 Mha and approaching zero in several months. During

## Burned area fraction 2019-2024

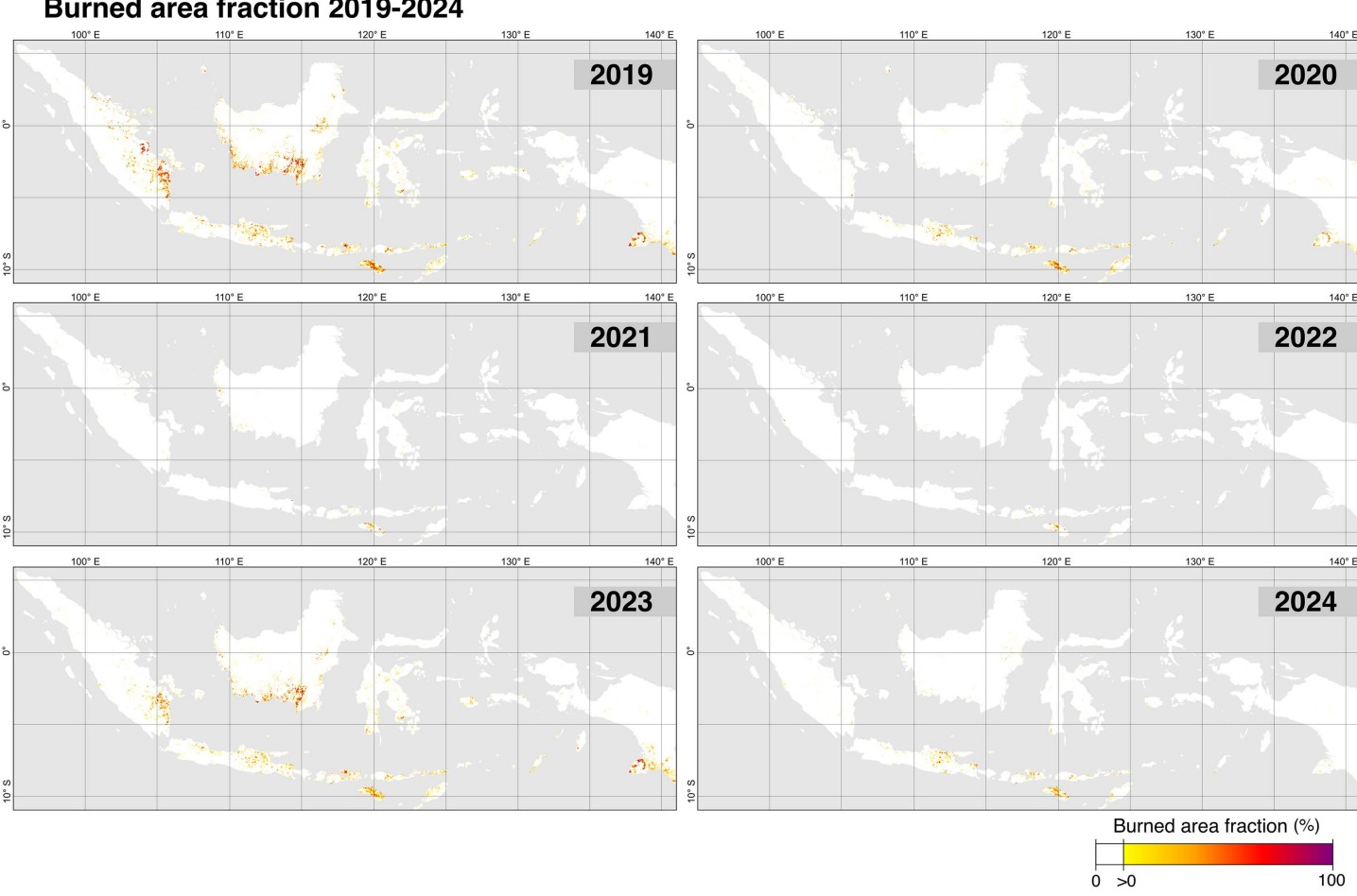

**Fig 7. Spatial distribution of the burned area fraction (2019–2024) across Indonesia, mapped at a 1 km resolution.** Each panel represents the burned area fraction for a specific year, calculated as the proportion of the pixel area affected by fire during that year. The color gradient from yellow to red indicates the burned area fraction, ranging from low (0%) to high (100%). The maps reveal significant spatial and interannual variability in burned area patterns. For this figure, the 20-m burned area layer was aggregated to a 1-km burned area fraction to allow the visualization at the country level. A 20-m map has too much detail to be clearly shown over large areas.

positive IOD and El Niño phases, reduced convection and rainfall lead to drier and warmer conditions across much of Indonesia, increasing fuel flammability and extending the dry season. Conversely, La Niña and negative IOD phases enhance rainfall and humidity, reducing fire occurrence. These results highlight the critical role of El Niño and positive IOD events in driving severe fire seasons in Indonesia, whereas neutral and La Niña conditions are associated with substantially reduced burned area.

## Accuracy assessment

In the seven provinces for which we assessed accuracy, our monthly burned area estimate, aggregated over one year (2019) presents a high user's accuracy (UA) for the 'burned' class, at 98.7% (CI: 97.7%–99.6%) indicating a low 1.3% (CI: 0.4%−2.3%) commission-error rate (Table 1). This result is on par with our previously published Annual estimate, which reported a similar UA for the 'burned' class, at a 97.9% (CI: 97.1%−98.8%) [30]. By contrast, the MCD64A1 data had a much lower UA for the burned class, at 76.0% (CI: 73.3%−78.7%).

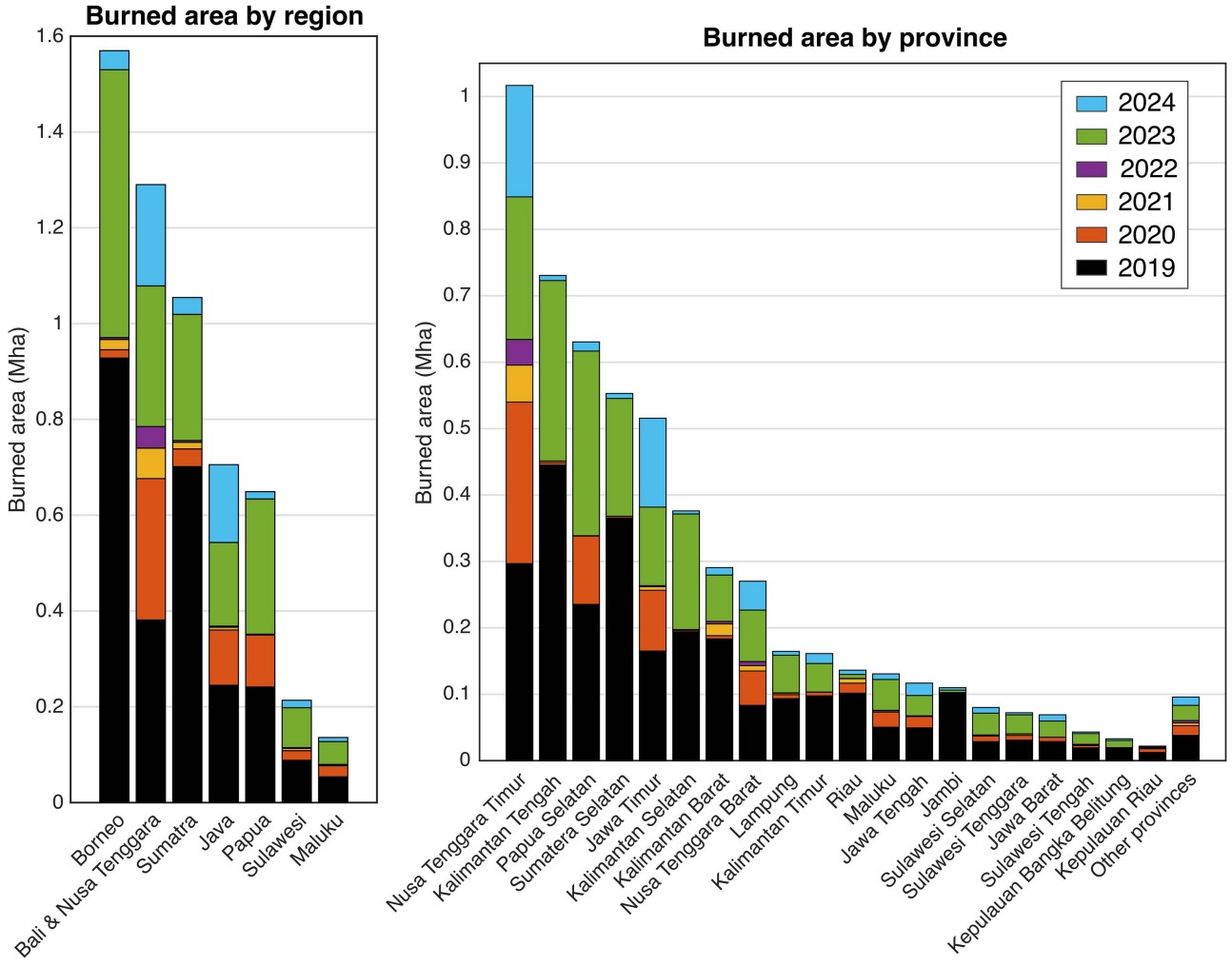

**Fig 8. Total burned area by region (left) and province (right) in Indonesia from 2019 to 2024.** Burned area is shown in million hectares (Mha), with bars stacked by year to illustrate interannual variability. The highest levels of fire activity occurred in Kalimantan, Bali & Nusa Tenggara, and Sumatra, with the provinces of Nusa Tenggara Timur, Kalimantan Tengah, Papua Selatan, and Sumatera Selatan accounting for 52.2% of the national total across the six-year period. Peak fire years (2019 and 2023) dominate most provinces, while lower activity is seen in 2021 and 2022.

The producer's accuracy (PA) for the 'burned' class in our Monthly estimate was 65.3% (CI: 57.9%–72.7%). This PA is lower than the metric reported for the Annual burned-area estimate, at 75.6% (CI: 68.3%–83.0%), although the difference is not statistically significant. This translates to an omission error rate of 34.7% (CI: 27.3%–42.1%) for the Monthly dataset. While the confidence intervals overlap slightly, our monthly approach shows a higher PA than the MCD64A1 product, which reported a PA of 53.1% (CI: 45.8%–60.5%), suggesting improved sensitivity to burned areas. This highlights that while the Monthly burned-area product has a higher omission error than the annual product, it still offers a notably improved detection rate compared to MCD64A1. Both burned-area datasets underestimate the true burned area extent, as per their respective PA figures, but our Monthly burned-area processing chain strikes a stronger balance between producer and user accuracy, while delivering timely, monthly updates at fine spatial resolution.

The confusion matrix (Fig 11) shows the count of reference sites (n = 259) classified as burned in a given month by the monthly processing chain (y-axis) and the month in which the same sites were confirmed as burned by visual inspection

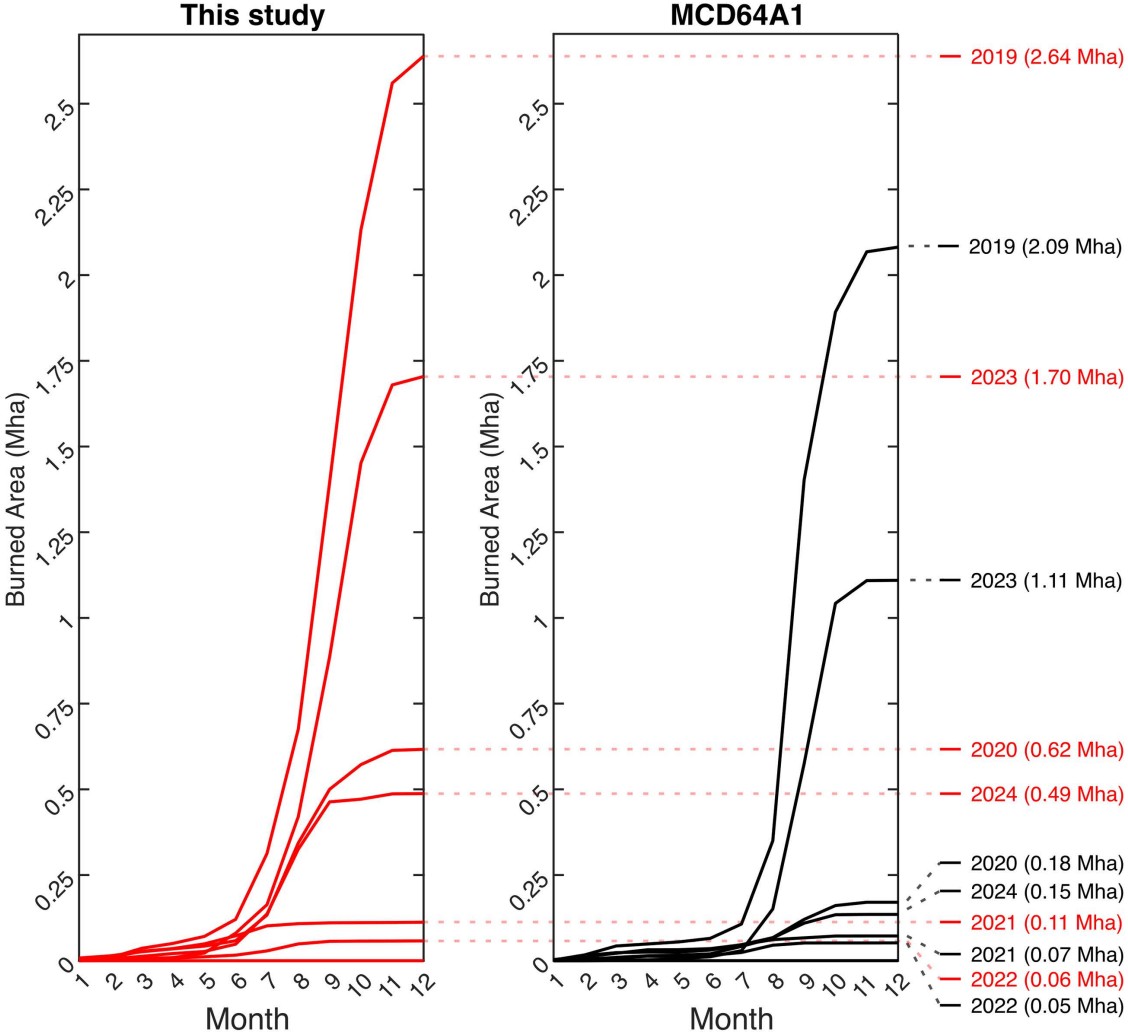

**Fig 9. Indonesia-wide cumulative burned area (in million hectares, Mha) by month for years 2019 to 2024.** This graph compares results from this study (left panel, red lines) and the MCD64A1 burned area product (right panel, black lines). Each line represents the cumulative burned area for a given year, with final annual totals indicated on the right. The sharp increase in burned area from June to October reflects the seasonal pattern typical of fire activity in Indonesia, driven by dry conditions during these months. The consistent underestimation by MCD64A1 highlights the improved detection capacity of this study's approach.

(x-axis) in the original point dataset. The results demonstrate strong temporal alignment, with 206 out of 250 sites (80%) correctly assigned. 68 sites were correctly assigned to September and 56 sites to October — the two peak fire months in 2019. Only 2% of the points showed detection mismatches longer than one month. This indicates that our method assigns fire timing accurately in most cases, while also reflecting the known difficulty of detecting fires that span multiple months.

## Burn size comparison

To assess how fires of different sizes contribute to total burned area, we examined the cumulative burned area as a function of burned patch size for each year from 2019 to 2024 (Fig 12). Burned patch sizes are presented on a logarithmic scale to accommodate the wide range observed, from small burn patches of just a few hectares to very large scars spanning tens of

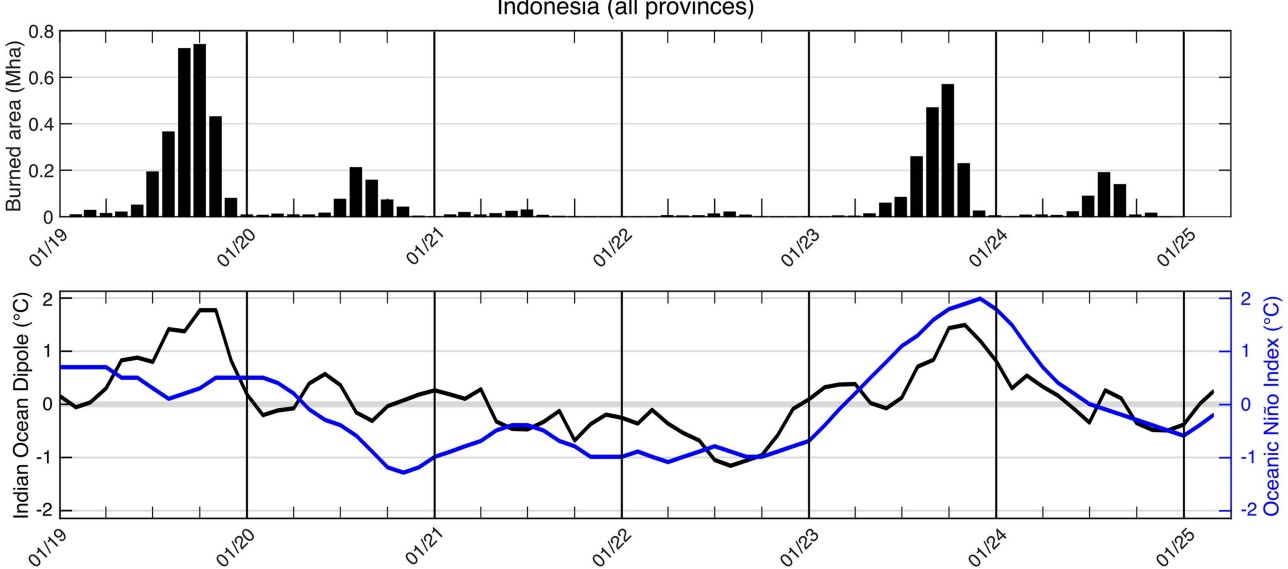

**Fig 10. Monthly burned area (top panel) and climate indices (bottom panel) for Indonesia from January 2019 to December 2024.** The top panel shows total burned area across all provinces (in million hectares, Mha). The bottom panel displays the Indian Ocean Dipole (IOD; black line, left axis, °C) and Oceanic Niño Index (ONI; blue line, right axis, °C), representing major climate anomalies. Peaks in burned area during late 2019 and mid to late 2023 align with positive IOD and El Niño conditions, while reduced burned area between 2020 and 2022 corresponds to neutral or negative phases of both indices. Vertical lines mark the start of each calendar year.

**Table 1. Accuracy assessment of each of the three burned-area maps performed in seven Indonesian provinces (87.60 Mha) targeted for peatland restoration. The accuracy metrics were estimated with a 1,042 points randomly distributed using stratified sampling. The reported metrics are (1) the overall accuracy (OA), the user accuracy (UA), and the producer accuracy (PA) with their 95 % confidence intervals.**

|  |  | Annual (Gaveau et al. 2021) | Monthly (This study) | MCD64A1 |
|---|---|---|---|---|
| OA (%) |  | 99.3 (99.0, 99.6) | 99.0 (98.7, 99.3) | 98.4 (98.0, 98.7) |
|  | Unburned | 99.3 (99.1, 99.6) | 99.0 (98.7, 99.3) | 98.8 (98.4, 99.1) |
| UA (%) | Burned | 97.9 (97.0, 98.7) | 98.7 (97.7, 99.6) | 76.0 (73.3, 78.7) |
|  | Unburned | 100.0 (99.3, 100.0) | 100.0 (99.0, 100.0) | 99.6 (98.7, 99.6) |
| PA (%) | Burned | 75.6 (68.3, 83.0) | 65.3 (57.9, 72.7) | 53.1 (45.8, 60.5) |

thousands of hectares. The results reveal consistent differences between our monthly burned area maps (Sentinel-2, this study) and those from the MCD64A1 product. Across all years, our dataset (red line) captures a larger cumulative burned area across the full spectrum of patch sizes. This discrepancy is especially marked in major fire years like 2019 and 2023, where our processing chain detects many larger, burned patches (>1,000 ha) than MCD64A1. Even in low-fire years such as 2021 and 2022, our approach identifies more burned area in small patches (<100 ha), underscoring MCD64A1's limited ability to detect fine-scale fires. Notably, the red curves from our maps show a near-linear trend on the log-scale axis, indicating a size-frequency pattern that approximates a power-law distribution — characteristic of scale-invariant fire regimes. In contrast, the MCD64A1 curves are more irregular and consistently fall below ours, confirming their systematic underestimation of burned area, particularly for small and fragmented fires. These results highlight the value of high-resolution, monthly Sentinel-2 data for producing more complete and accurate burned area assessments across Indonesia.

The statistical results in Table 2 support these findings. We applied the Kruskal-Wallis test to compare scar size frequency distributions across datasets at four patch size thresholds (25, 100, 1,000, and 5,000 ha). At

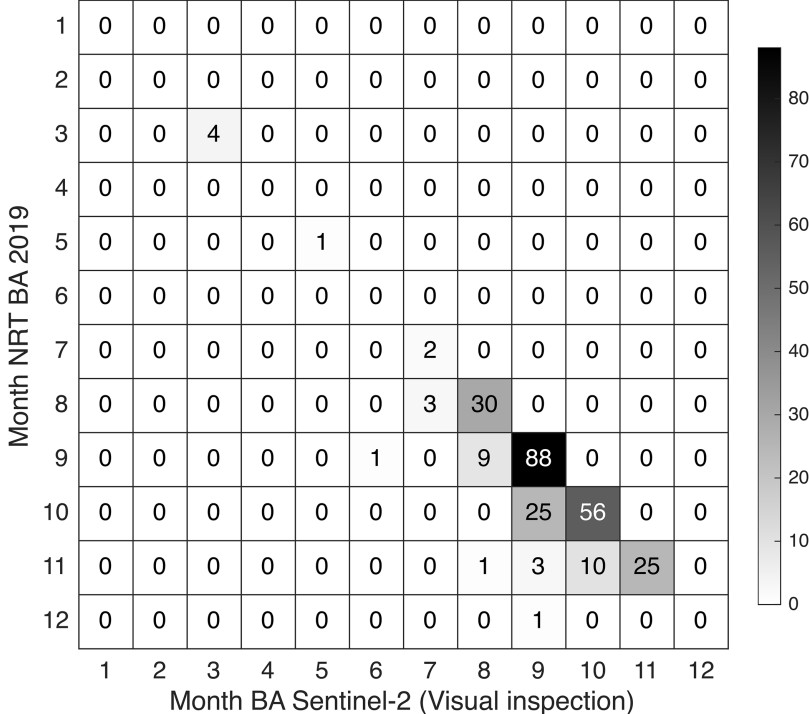

**Fig 11. Confusion matrix of burn-month detection from processing chain versus visual interpretation.** This matrix compares the month of detection assigned by the monthly burned-area processing chain (NRT, y-axis) with the burn month determined by visual interpretation of Sentinel-2 imagery (x-axis) for 259 reference sites in 2019. The matrix shows the number of sites detected as "burned" in each month by the processing chain versus the month confirmed through visual inspection. High counts along the diagonal indicate strong agreement between the processing chain and visual interpretation. Off-diagonal values reflect either persistent visibility of burn scars or timing discrepancies of detection.

fine resolutions (25 and 100 ha), p-values are extremely low ($< 10^{-24}$), indicating significant differences in the size distributions among Sentinel-2, MCD64A1, and official maps. This confirms that the products diverge most strongly at small patch sizes, where MCD64A1 consistently underrepresents burned areas. In contrast, at larger size thresholds (1,000 and 5,000 ha), the p-values exceed 0.3 and 0.6, respectively, showing no significant differences.

## Discussion

In Indonesia, fire is both a farming tool and a natural feature of savanna ecosystems, but it can also become a destructive force that degrades humid forests, swamps, and agricultural lands. Government agencies already monitor fire activity every month [23], with some success, but current systems rely heavily on visual interpretation of Landsat, and on FIRMS hotspots, which makes it difficult to capture the complete extent of fire-affected areas.

This study presents the first operational monthly burned-area processing chain designed to detect fires in unprecedented detail for Indonesia. The system combines high-frequency Sentinel-2 imagery (every 3–5 days) with FIRMS daily fire hotspots to produce 20-m resolution monthly burned-area maps. The processing chain runs in Google Earth Engine and delivers timely, monthly information that can support emergency response, forensic investigations, law enforcement, and conservation planning across Indonesia.

From 2019 to 2024, our processing chain mapped 5.62 million hectares of cumulative burned area, equivalent in size to the country of Croatia. Most fires occurred in Kalimantan, Nusa Tenggara, and Sumatra. Six of Indonesia's 38 provinces

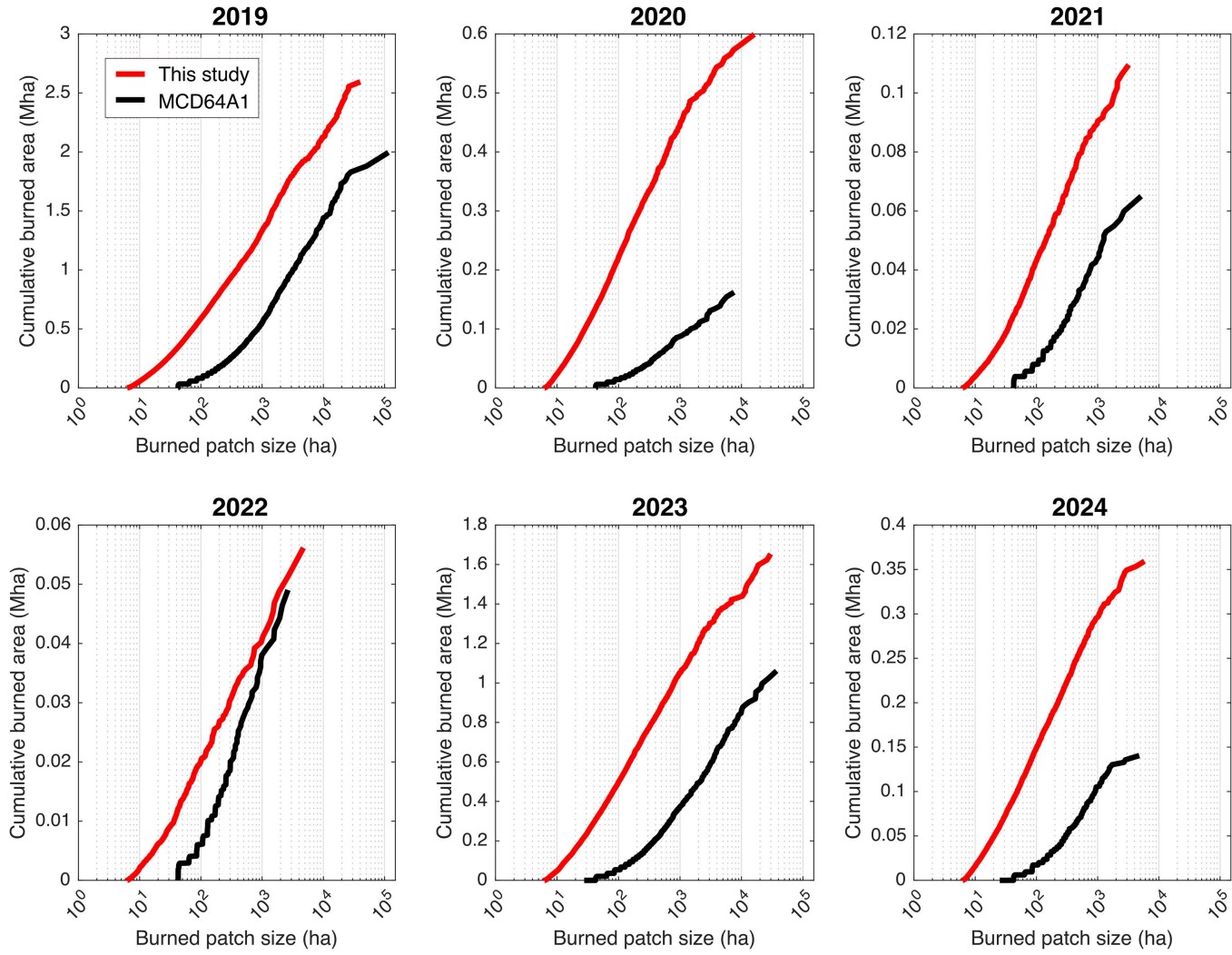

**Fig 12. Cumulative burned area as a function of burned patch size for each year from 2019 to 2024.** The red line shows results from this study, while the black line represents the MODIS MCD64A1 burned area product. For each year, cumulative burned area (in million hectares, Mha) is plotted against individual burned patch sizes (in hectares, ha, logarithmic scale). The x-axis represents burned patch size on a logarithmic scale, allowing for a clear visualization of both small and large patches, which vary over several orders of magnitude. Across all years, this study consistently maps a larger cumulative burned area and detects more burned patches across a wide range of sizes compared to MCD64A1. The differences are especially pronounced for large patches in major fire years (2019 and 2023), but are also evident for small, burned patches, particularly in low-fire years (2020–2022), indicating that MCD64A1 underestimates both small and large fires in Indonesia.

**Table 2. Tests statistics with respect to differences in burned-area scar size frequency distributions for Sentinel, MODIS, and official maps.**

| Scar size (ha) | Kruskal-Wallis test (p-values) |
|---|---|
| 25 | 2.9537e-291 |
| 100 | 3.8813e-25 |
| 1000 | 0.32109 |
| 5000 | 0.65824 |

 

— Nusa Tenggara Timur, Kalimantan Tengah, Papua Selatan, Sumatra Selatan, Jawa Timur, and Kalimantan Selatan — accounted for 68% of the total burned area. These regions reflect three main fire regimes, i) large wildfires on degraded previously deforested lands, including peatlands in Kalimantan Tengah, Sumatra Selatan, and Kalimantan Selatan, ii) frequent savanna wildfires in Papua Selatan and NTT islands such as Sumba, and iii) frequent maintenance agricultural fires in Jawa Timur. Among these fire regions, savanna and maintenance fires pose comparatively lower environmental concerns.

We found that severe fire seasons (2019 and 2023) coincided with positive phases of both the Indian Ocean Dipole (IOD) and the El Niño–Southern Oscillation (ENSO), the latter measured by the Oceanic Niño Index (ONI). This pattern aligns with previous findings [7], which showed that the combined influence of ENSO and IOD intensify dry spells and fire activity across Sumatra and Kalimantan. Fire activity accelerated from July and peaked in September–October when ONI ≥ ~+0.5 °C and IOD ≥ +1.5 °C. In contrast, 2020–2022 saw minimal burning under neutral or negative phases of both indices, while 2024 showed moderate fire activity without strong anomalies. These findings reaffirm the importance of tracking both indices (ONI and IOD) for early warning. We note that this analysis is descriptive and intended to illustrate temporal co-variation between climatic indices and monthly burned-area fluctuations, rather than to quantify causality. Future work will use statistical models and incorporate additional climatic variables such as air temperature, rainfall, vapor pressure deficit, or climatic water deficit. These variables are known drivers of fire occurrence [42] that can determine the magnitude and causality of climatic influences.

The Indonesian government has reduced forest fires in recent years [43], and our analysis confirms this trend. The last two fire seasons in which both the El Niño–Southern Oscillation (ENSO) and the Indian Ocean Dipole (IOD) were in positive phases (2015 and 2023) produced comparable drought across Indonesia. The 2019 fire season, although not linked to El Niño, was driven by an exceptionally strong positive IOD, the most intense since the 1960s, which also caused severe rainfall deficits in Kalimantan and Sumatra [44,45]. Despite the extreme dry conditions, we found that in both 2019 and 2023, 122,164 hectares of primary humid forest burned, equivalent to 2.2% of the total burned area. This contrasts sharply with previous extreme fire years: in 1997/98, an estimated 5 million hectares of forest burned [12], while in 2015/16, the figure was 450,000 hectares [46]. These findings suggest that recent fire prevention and response efforts have a tangible impact, particularly in humid forests during severe drought years.

Our monthly Sentinel-2 approach consistently detected more burned area compared to the widely used MCD64A1 product. This discrepancy is especially marked in major fire years like 2019 and 2023, when our processing chain detected larger, burned patches (>1,000 ha). Even in low-fire years such as 2021 and 2022, our approach identifies more burned area in small patches (<100 ha), which highlight the inability of MCD64A1 to detect small fires compared to our results. This underestimation is expected because the 500-m MCD64A1 product cannot reliably detect smaller fires [17]. Similar conclusions were reached by over Sub-Sahara Africa [20] when comparing the MCD64A1 product with 'Small Fire Dataset' derived from Sentinel-2 [47]. Our burn-size analysis showed that Sentinel-2 results follow a scale-free distribution [48,49], which is common of wildfire regimes worldwide [50]. In contrast, MCD64A1 showed an S-shaped curve that indicated bias at both small and large scales. Our results further demonstrates that coarse-resolution products underestimate both the smallest and largest burn scars. Accurate detection of both ends of this size spectrum is important for characterizing fire regimes (wildfire, agricultural fire), their scale and severity. Despite this, MCD64A1 remains a valuable globally operational product that provides consistent long-term burned area records across the world. Understanding how it compares with finer-resolution datasets such as Sentinel-2 is useful for bias correction and better interpretation of global fire trends.

Validation confirmed the reliability of our processing chain, with a user's accuracy for the "burned" class of 98.7% and a commission error of only 1.3% (CI: 0.4%–2.3%). The producer's accuracy of 65.3% and omission error of 34.7% (CI: 27.3%–42.1%) reflect our conservative approach that minimizes false positives. We prioritised a low commission error rate (i.e., high user accuracy) to address sensitivities [51]. This makes the maps suitable for legal and enforcement use.

The omission of burn scars commonly occur in low-intensity fires beneath forest canopies [52], short-lived fires in savannas grassland that re-green within a week, or burn scars that are obscured by persistent cloud or haze. Although our approach produces conservative burned area estimates, it ensures that every detected scar can be used with high confidence, particularly when overlaid with contextual layers such as land zoning plans. For instance, the overlay of our burned area maps with concession boundaries allows authorities to produce objective fire evidence every month. This can help to act quickly, impose fines, and hold landowners accountable.

The proportion of burned area occurring on peatlands in our analysis (16.1%) is lower than that reported by the MCD64A1 burned-area product (27.4%) and lower than our own previous estimates for year 2019 [30]. This is because we used an updated national peatland map from the Indonesian Ministry of Agriculture, which has a smaller spatial extent than earlier peatland datasets [53]. These differences show that estimates of peatland area burning are sensitive to how peatland extent is mapped. They emphasize the importance of continued improvements in national peatland maps given that peat fires reduction is central to addressing concerns about regional air quality and achieving Indonesia's ambitious national carbon-emissions reduction target.

This study presents the first operational method that produces monthly 20-m resolution burned-area maps for Indonesia. The workflow is largely automated and fully replicable, and provides high-confidence data suitable for consistent national monitoring. To support wider adoption, the burned-area maps and their monthly updates have been integrated into the Nusantara Atlas platform (www.nusantara-atlas.org), an open-access platform for monitoring deforestation in Southeast Asia. This supports month-by-month burned-area estimates across a broad array of land units, including oil palm, pulpwood and mining concessions, administrative units (down to the municipality level), and protected areas, for use by government agencies, NGOs, and the public. The operational execution, automation, and integration of the processing chain with the Nusantara Atlas platform are described in Supplementary Methods. Long-term availability of the data will depend on future cloud-processing and data-hosting costs under Google's pricing policy. Overall, the system provides a robust and replicable framework for operational burned-area monitoring in Indonesia and can be adapted to other tropical regions facing similar challenges.

## Supporting information

**S1 Text. Processing pipeline for Nusantara Atlas.** Description of the processing pipeline used to integrate the results into the Nusantara Atlas platform.
(DOCX)

## Author contributions

**Conceptualization:** David L.A. Gaveau.

**Data curation:** Mohammad Agus Salim.

**Formal analysis:** David L.A. Gaveau, Adrià Descals.

**Investigation:** David L.A. Gaveau.

**Methodology:** David L.A. Gaveau, Adrià Descals.

**Software:** Mohammad Agus Salim, Adrià Descals.

**Validation:** David L.A. Gaveau, Adrià Descals.

**Visualization:** Mohammad Agus Salim, Adrià Descals.

**Writing – original draft:** David L.A. Gaveau, Adrià Descals.

**Writing – review & editing:** David L.A. Gaveau, Adrià Descals, Mohammad Agus Salim.

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
