## [Decision Letter · Decision Letter 0]

22 Oct 2025

PONE-D-25-45724Nationwide Monthly Burned Area monitoring in Indonesia using Sentinel-2PLOS ONE

Dear Dr. Gaveau,

Thank you for submitting your manuscript to PLOS ONE. After careful consideration, we feel that it has merit but does not fully meet PLOS ONE’s publication criteria as it currently stands. Therefore, we invite you to submit a revised version of the manuscript that addresses the points raised during the review process.

We look forward to receiving your revised manuscript.

Kind regards,

Bijeesh Kozhikkodan Veettil

Academic Editor

PLOS ONE

Journal Requirements:

4. We note that Figures 3, 4, 5, and 8 in your submission contain map/satellite images which may be copyrighted. All PLOS content is published under the Creative Commons Attribution License (CC BY 4.0), which means that the manuscript, images, and Supporting Information files will be freely available online, and any third party is permitted to access, download, copy, distribute, and use these materials in any way, even commercially, with proper attribution. For these reasons, we cannot publish previously copyrighted maps or satellite images created using proprietary data, such as Google software (Google Maps, Street View, and Earth). For more information, see our copyright guidelines: http://journals.plos.org/plosone/s/licenses-and-copyright.

1. You may seek permission from the original copyright holder of Figures 3, 4, 5, and 8 to publish the content specifically under the CC BY 4.0 license.

Reviewers' comments:

Reviewer's Responses to Questions

**Comments to the Author**

1. Is the manuscript technically sound, and do the data support the conclusions?

Reviewer #1: Partly

Reviewer #2: Yes

2. Has the statistical analysis been performed appropriately and rigorously? 

Reviewer #1: No

Reviewer #2: Yes

3. Have the authors made all data underlying the findings in their manuscript fully available?

The PLOS Data policy requires authors to make all data underlying the findings described in their manuscript fully available without restriction, with rare exception (please refer to the Data Availability Statement in the manuscript PDF file). The data should be provided as part of the manuscript or its supporting information, or deposited to a public repository. For example, in addition to summary statistics, the data points behind means, medians and variance measures should be available. If there are restrictions on publicly sharing data—e.g. participant privacy or use of data from a third party—those must be specified.requires authors to make all data underlying the findings described in their manuscript fully available without restriction, with rare exception (please refer to the Data Availability Statement in the manuscript PDF file). The data should be provided as part of the manuscript or its supporting information, or deposited to a public repository. For example, in addition to summary statistics, the data points behind means, medians and variance measures should be available. If there are restrictions on publicly sharing data—e.g. participant privacy or use of data from a third party—those must be specified.requires authors to make all data underlying the findings described in their manuscript fully available without restriction, with rare exception (please refer to the Data Availability Statement in the manuscript PDF file). The data should be provided as part of the manuscript or its supporting information, or deposited to a public repository. For example, in addition to summary statistics, the data points behind means, medians and variance measures should be available. If there are restrictions on publicly sharing data—e.g. participant privacy or use of data from a third party—those must be specified.requires authors to make all data underlying the findings described in their manuscript fully available without restriction, with rare exception (please refer to the Data Availability Statement in the manuscript PDF file). The data should be provided as part of the manuscript or its supporting information, or deposited to a public repository. For example, in addition to summary statistics, the data points behind means, medians and variance measures should be available. If there are restrictions on publicly sharing data—e.g. participant privacy or use of data from a third party—those must be specified.

Reviewer #1: No

Reviewer #2: Yes

4. Is the manuscript presented in an intelligible fashion and written in standard English?

Reviewer #1: Yes

Reviewer #2: Yes

5. Review Comments to the Author

Reviewer #1: 1.In remote sensing and computer science, algorithm often implies a novel mathematical or computational procedure.

In this manuscript, the approach is essentially a workflow combining preprocessing, machine learning, and rule-based filtering. The claim of developing a new “algorithm” is overstated, as most components rely on established methods and primarily represent an operational integration of existing techniques. A more accurate framing might be processing chain or workflow/ generic methodology rather than a novel algorithm.

2.If the authors claim this is the first automated nationwide monthly burned area algorithm, reviewers and readers will expect at least a description of how end users can interact with it. Without an interface or demonstration, the system risks being perceived as a research prototype rather than an operational monitoring tool. This is especially critical because the paper emphasizes policy and enforcement applications

3.The link between the ENSO (ONI) and IOD anomalies and fire peaks is described as correlation, but no statistical modeling is provided to quantify strength of influence.

Reviewer #2: The paper is well-written in a consistent manner and is easy to understand. The topic is interesting and very important for conserving forests and vegetation cover in tropical areas. I only have some recommendations, which I have provided below:

The abstract does not mention the methodology.

Check the text for language issues.

Line 67: Provide some references regarding the use of Landsat data in Indonesia.

Line 81: "with advanced machine learning algorithms" – State exactly which methods were used.

Line 91: "To date, no automated burned area detection system exists for Indonesia." I did a brief search and found several papers using AI for burnt area detection. How can the authors claim that their method is the only one published so far? I think there are some other papers published, using AI as an automated way of detecting burned areas in Indonesia.

Line 94: Explain the "FIRMS" dataset, as it might be new to some readers.

What is the difference between this paper and the previous one? (See line 79).

Figure 1 is very simple. The paper is fine without it as it is. Consider adding more detail or removing it, as the text is already sufficient.

The text uses many abbreviations without explaining them fully.

I don't understand why 6.25 hectares was selected as the threshold. This is already a large area (250m x 250m). The authors have stated that not removing fragmented burned areas is a main advantage of this paper. However, many of these areas could be false positives or negatives. What is the impact on the final results if these patches are removed? Is this done only for comparison with other products?

In section 3.2, please explain what happens in the different phases of each oscillation regarding precipitation, temperature, etc., and how these factors affect burned areas.

6. PLOS authors have the option to publish the peer review history of their article (what does this mean?). If published, this will include your full peer review and any attached files.). If published, this will include your full peer review and any attached files.). If published, this will include your full peer review and any attached files.). If published, this will include your full peer review and any attached files.

...

Reviewer #1: No

Reviewer #2: **Yes:**Masoud Jafari ShalamzariMasoud Jafari ShalamzariMasoud Jafari ShalamzariMasoud Jafari Shalamzari

---

## [Author Response · Author response to Decision Letter 1]

29 Dec 2025

Response to Reviewers

Manuscript title: Automated monthly burned area mapping for Indonesia using Sentinel-2 and FIRMS data (2019–2024)

Manuscript ID: PONE-S-25-59671

Authors: D.L.A. Gaveau, A. Descals, M.A. Salim

Journal: PLOS ONE

General Response

We thank the Academic Editor and both reviewers for their constructive and insightful comments. We have revised the manuscript accordingly. Below we address each comment point-by-point. Reviewer comments are reproduced in italics, followed by our responses in plain text.

Reviewer #1

“In remote sensing and computer science, algorithm often implies a novel mathematical or computational procedure. In this manuscript, the approach is essentially a workflow combining preprocessing, machine learning, and rule-burned-area filtering… The claim of developing a new ‘algorithm’ is overstated.”

Response:

We appreciate this clarification. We agree that the term workflow or processing chain more accurately describes our method. We have revised the manuscript throughout to reduce the emphasis on “new algorithm” and instead describe it as an “automated monthly burned-area processing chain” built from established methods — Sentinel-2 pre/post-fire compositing, Random Forest classification, and rule-burned-area filtering — integrated into a single operational workflow.

The novelty lies in (i) its automation at national scale, (ii) its monthly temporal resolution, and (iii) its adaptation to Indonesia’s landscape and policy requirements.

“If the authors claim this is the first automated nationwide monthly burned area algorithm, readers will expect at least a description of how end users can interact with it… Without an interface or demonstration, the system risks being perceived as a research prototype.”

Response:

We agree and have clarified in the revised manuscript how the system will be made accessible to end users. The automated workflow will being integrated into the Nusantara Atlas platform (www.nusantara-atlas.org), which already provides near-real-time deforestation and fire-alert data. We will publish the monthly burned-area layers on Nusantara Atlas. This will ensure operational accessibility for the public, NGOs and relevant agencies. To reflect this, the Abstract now includes:

“Monthly burn-scar updates (post-December 2024 to present) are available for viewing and analysis via Nusantara Atlas (www.nusantara-atlas.org), where the system operates as an open, operational platform for public use..”

The Discussion also now concludes with a paragraph explaining the implementation and access considerations in full detail: “To support wider adoption, we have integrated our burned-area maps into the Nusantara Atlas platform (www.nusantara-atlas.org), which already provides near-real-time deforestation and fire hotspots data. This integration will enable continuous updates of monthly burned-area maps and make them accessible for analysis by government agencies, NGOs, and the public.

“The link between ENSO (ONI) and IOD anomalies and fire peaks is described as correlation, but no statistical modeling is provided to quantify strength of influence.”

Response:

We thank the reviewer for this comment. We compared the ENSO (ONI) and IOD anomalies with the burned area monthly as an additional validation exercise to verify that the burned area seasonality aligns with the expected fire season patterns linked with ENSO and IOD and reported in previous studies. Our goal was not to model the statistical strength of influence but to confirm that the temporal variations in burned area are consistent with known large-scale climate drivers. We have clarified this point in the Discussion section. The revised text now explicitly notes that the analysis is descriptive.

We wrote: “We note that this analysis is descriptive and intended to illustrate temporal co-variation between climatic indices and monthly burned-area fluctuations, rather than to quantify causality. Future work will apply statistical modelling approaches incorporating additional climatic variables such as air temperature, rainfall, vapor pressure deficit, or climatic water deficit, known drivers of fire occurrence,to estimate the strength and causality of these climatic influences.”

Reviewer #2

“The abstract does not mention the methodology.”

Response:

We have revised the abstract to include a concise description of the method used to map burned areas:

“Our approach uses a Random Forest model that classifies Sentinel-2 imagery and uses FIRMS fire hotspots to reduce false positives. The resulting 20-m monthly burned-area maps cover the entire country from January 2019 to December 2024.”

“Check the text for language issues.”

Response:

The manuscript has been carefully read to correct minor grammatical and typographical issues, ensuring clear and standard English throughout.

“Line 67: Provide some references regarding the use of Landsat data in Indonesia.”

Response:

We have added the following reference to support the statement about Landsat’s use for burned-area mapping in Indonesia by the Indonesian government:

https://sipongi.gakkum.kehutanan.go.id/indikasi-luas-kebakaran

In this reference, it is written: “The area of forest and land fires is calculated burned-areas on Landsat-8 OLI/TIRS image analysis”

“Line 81: ‘with advanced machine learning algorithms’ – State exactly which methods were used.”

Response:

We have clarified that the workflow uses the Random Forest classifier implemented in Google Earth Engine. The sentence now reads: “a method that combines time-series Sentinel-2 imagery with a machine learning classification model (Random Forest), implemented within the Google Earth Engine.”

“Line 91: ‘To date, no automated burned area detection system exists for Indonesia.’ I found several papers using AI for burnt area detection… How can the authors claim that their method is the only one?”

Response:

We appreciate this correction; and have revised the sentence for accuracy.

We now state: “While previous studies have proposed promising methodologies for monthly burned-area mapping in Indonesia, using Sentinel-2 and other remote sensing inputs (Vetrita et al., 2025; Arjasakusuma et al., 2022), none have yet delivered an automated, nationwide, monthly burned-area mapping system that operates continuously.”

This distinguishes our contribution in terms of spatial coverage, temporal frequency, and operational automation.

“Line 94: Explain the ‘FIRMS’ dataset, as it might be new to some readers.”

Response:

We have added the following text that explains the FIRMS dataset:

We now say in the Introduction: “Our method integrates high-frequency Sentinel-2 time series (every 2–5 days) with FIRMS (Fire Information for Resource Management System) daily fire alerts (Giglio et al., 2016), which are derived from MODIS (Moderate Resolution Imaging Spectroradiometer) sensors onboard NASA’s Terra and Aqua satellites and the VIIRS (Visible Infrared Imaging Radiometer Suite) sensors onboard the Suomi NPP and NOAA-20 satellites (NASA, 2025). These instruments detect daily thermal infrared anomalies associated with active combustion. In this study, the resulting fire hotspot detections serve as both spatial and temporal filters for identifying candidate burned areas detected and delineated using Sentinel-2 imagery.”

“What is the difference between this paper and the previous one (line 79)?”

Response:

We now clarify this distinction in the Introduction section:

“Our earlier study (Gaveau et al., 2021) produced an annual burned-area product suitable for retrospective analysis, whereas the present study develops a monthly, automated system designed for near-real-time monitoring, enabling continuous national updates.”

“Figure 1 is very simple. The paper is fine without it as it is. Consider adding more detail or removing it.”

Response:

We have removed Figure 1.

“The text uses many abbreviations without explaining them fully.”

Response:

We have ensured that all abbreviations (e.g., GEE, FIRMS, NBR, ONI, IOD, etc.) are defined upon first use in the manuscript.

We replace the BA abbreviation with “burned-area”

“I don't understand why 6.25 hectares was selected as the threshold…”

Response:

We thank the reviewer for raising this point.

Our classification system can detect burn scars smaller than 6.25 ha; however, we adopted this threshold to align with the official minimum mapping unit (MMU) used by the Indonesian Ministry of Forestry for national burned-area assessments.

This ensures comparability with official datasets and legal frameworks. Smaller patches are filtered from the published layer to meet this national standard.

The revised manuscript now makes this clear in the Introduction section: “ The workflow generates monthly burn-scar maps at 20-meter resolution, with a minimum mapping unit of 6.25 hectares — aligned with the official minimum burn-scar size recognized by Indonesian authorities”

(MoEF): Indonesia’s Second Forest Reference Emission Level (FREL) Submission to the UNFCCC, Ministry of Environment and Forestry (MoEF), Jakarta, Indonesia, 2022. https://redd.unfccc.int/files/modified_2nd_frl_indonesia_20220529_clean.pdf

“In section 3.2, please explain what happens in the different phases of each oscillation regarding precipitation, temperature, etc., and how these factors affect burned areas.”

Response:

We have added a paragraph in Section 3.2 explaining why positive IOD and El Niño (ONI) phases typically increase fire risk, while negative phases reduce it.

We wrote in Section 3.2: “During positive IOD and El Niño phases, reduced convection and rainfall lead to drier and warmer conditions across much of Indonesia, increasing fuel flammability and extending the dry season. Conversely, La Niña and negative IOD phases enhance rainfall and humidity, reducing fire occurrence.”

This provides the climatic context for the observed burned-area variations. We note in the Discussion section that future work will apply statistical modelling approaches incorporating additional climatic variables such as air temperature, rainfall, vapor pressure deficit, or climatic water deficit, known drivers of fire occurrence (Descals et al., 2022), to estimate the causality of these climatic influences.

Data Availability

Reviewer #1 expressed concern that data were not fully available.

We have clarified in the Data Availability Statement that:

• All input datasets (Sentinel-2 SR, FIRMS, MODIS MCD64A1) are publicly accessible.

• The monthly burned-area outputs will be hosted openly via the Nusantara Atlas upon acceptance. Thus, we will include the link to the dataset in the accepted version.

We believe these revisions address all reviewer concerns and substantially improve the manuscript’s clarity, rigor, and accessibility.

---

## [Decision Letter · Decision Letter 1]

22 Jan 2026

PONE-D-25-45724R1Nationwide Monthly Burned Area monitoring in Indonesia using Sentinel-2PLOS One

Dear Dr. Gaveau,

Thank you for submitting your manuscript to PLOS ONE. After careful consideration, we feel that it has merit but does not fully meet PLOS ONE’s publication criteria as it currently stands. Therefore, we invite you to submit a revised version of the manuscript that addresses the points raised during the review process.

We look forward to receiving your revised manuscript.

Kind regards,

Bijeesh Kozhikkodan Veettil

Academic Editor

PLOS One

Journal Requirements:

Reviewers' comments:

Reviewer's Responses to Questions

**Comments to the Author**

1. If the authors have adequately addressed your comments raised in a previous round of review and you feel that this manuscript is now acceptable for publication, you may indicate that here to bypass the “Comments to the Author” section, enter your conflict of interest statement in the “Confidential to Editor” section, and submit your "Accept" recommendation.

Reviewer #1: All comments have been addressed

2. Is the manuscript technically sound, and do the data support the conclusions?

Reviewer #1: No

3. Has the statistical analysis been performed appropriately and rigorously? 

Reviewer #1: Yes

4. Have the authors made all data underlying the findings in their manuscript fully available?

The PLOS Data policy requires authors to make all data underlying the findings described in their manuscript fully available without restriction, with rare exception (please refer to the Data Availability Statement in the manuscript PDF file). The data should be provided as part of the manuscript or its supporting information, or deposited to a public repository. For example, in addition to summary statistics, the data points behind means, medians and variance measures should be available. If there are restrictions on publicly sharing data—e.g. participant privacy or use of data from a third party—those must be specified.requires authors to make all data underlying the findings described in their manuscript fully available without restriction, with rare exception (please refer to the Data Availability Statement in the manuscript PDF file). The data should be provided as part of the manuscript or its supporting information, or deposited to a public repository. For example, in addition to summary statistics, the data points behind means, medians and variance measures should be available. If there are restrictions on publicly sharing data—e.g. participant privacy or use of data from a third party—those must be specified.requires authors to make all data underlying the findings described in their manuscript fully available without restriction, with rare exception (please refer to the Data Availability Statement in the manuscript PDF file). The data should be provided as part of the manuscript or its supporting information, or deposited to a public repository. For example, in addition to summary statistics, the data points behind means, medians and variance measures should be available. If there are restrictions on publicly sharing data—e.g. participant privacy or use of data from a third party—those must be specified.requires authors to make all data underlying the findings described in their manuscript fully available without restriction, with rare exception (please refer to the Data Availability Statement in the manuscript PDF file). The data should be provided as part of the manuscript or its supporting information, or deposited to a public repository. For example, in addition to summary statistics, the data points behind means, medians and variance measures should be available. If there are restrictions on publicly sharing data—e.g. participant privacy or use of data from a third party—those must be specified.

Reviewer #1: No

5. Is the manuscript presented in an intelligible fashion and written in standard English?

Reviewer #1: Yes

6. Review Comments to the Author

Reviewer #1: The authors provide a detailed narrative of analytical steps using Sentinel-2 compositing, Random Forest classification, post-classification filtering, validation, and dissemination via a web platform. However, these steps are described as analytical procedures rather than as components of an automated processing system.

Specifically:

a. The manuscript does not clearly describe whether the processing chain runs automatically on a fixed schedule.

b. How data ingestion, processing, quality control, and output publishing are orchestrated

c. Whether human intervention is required at any stage

d. No workflow diagram or system architecture figure is presented to illustrate,the interaction between platforms (Google Earth Engine, FIRMS, external climate datasets), and data flow from raw inputs to final products, and The interface between analysis and the Nusantara Atlas website

e. The manuscript suggests that burned-area results are “made available” through the Nusantara Atlas website, but it remains unclear whether the website is dynamically connected to the processing chain, or whether results are manually exported and uploaded.

f. The authors rely on several independent systems (Google Earth Engine, FIRMS, and a web visualization platform), yet the manuscript does not explain how these components are programmatically integrated, and whether a central controller or pipeline manages execution across platforms.

To substantiate the claim of automation, the authors should:

Explicitly describe the nature of the processing system

Include a workflow or system architecture diagram

Clearly distinguish between Analytical methods (classification, filtering, validation), and Operational automation (execution, updating, publishing).Without these additions, the manuscript primarily documents burned-area analysis and result dissemination via a website, rather than a demonstrably automated monitoring system.

7. PLOS authors have the option to publish the peer review history of their article (what does this mean?). If published, this will include your full peer review and any attached files.). If published, this will include your full peer review and any attached files.). If published, this will include your full peer review and any attached files.). If published, this will include your full peer review and any attached files.

...

Reviewer #1: No

---

## [Author Response · Author response to Decision Letter 2]

4 Mar 2026

Response to Reviewers

Manuscript title: Nationwide Monthly Burned Area Monitoring in Indonesia Using Sentinel-2

Manuscript ID: PONE-D-25-45724R1

Authors: D.L.A. Gaveau, A. Descals, M.A. Salim

Journal: PLOS ONE

General Response

We thank the Academic Editor and both reviewers for their constructive and insightful comments. We have revised the manuscript accordingly. Below we address each comment point-by-point. Reviewer comments are reproduced in italics, followed by our responses in plain text.

Reviewer #1

“The authors provide a detailed narrative of analytical steps… however these are described as analytical procedures rather than components of an automated processing system.”

Response:

We thank the reviewer for this insightful comment. We agree that the original manuscript did not sufficiently distinguish between the analytical methodology and the operational implementation of the processing chain. In response, we have revised the manuscript and added a dedicated description of the operational system as Supplementary Information.

We clarified the operational and automated nature of the processing chain. We added Supplementary Methods, describing execution, scheduling, orchestration, and system integration. We added Figure S1, illustrating the system architecture and data flow. We revised terminology throughout (e.g., “operational”, “largely automated”) for precision

“The manuscript does not clearly describe whether the processing chain runs automatically on a fixed schedule.”

Response:

We have now explicitly described the execution schedule in Supplementary Methods S1. The system operates on a fixed monthly schedule, automatically initiated on the 5th day of each month, allowing for Sentinel-2 data latency.

“How data ingestion, processing, quality control, and output publishing are orchestrated.”

Response:

Supplementary Methods S1 now describes the orchestration of:

• Automated Sentinel-2 data access within Google Earth Engine

• Automated FIRMS hotspot retrieval and ingestion

• Scripted GEE processing workflows

• Automated post-processing via Python scripts

• Dissemination through Nusantara Atlas

“ Whether human intervention is required at any stage.”

Response:

We have clarified that the workflow is largely automated, with human involvement limited to exception handling, such as system failures. No manual intervention is required during routine monthly execution.

“No workflow diagram or system architecture figure is presented”

Response:

We have added Figure S1, which presents a schematic of:

Data sources (Sentinel-2, FIRMS)

Google Earth Engine processing

Cloud storage

Post-processing

Nusantara Atlas dissemination

“ The manuscript suggests that burned-area results are “made available” through the Nusantara Atlas website, but it remains unclear whether the website is dynamically connected… or manually uploaded.”

Supplementary Methods S1 now clarifies that burned-area outputs are:

Exported automatically from GEE to Google Cloud Storage

Accessed programmatically by Nusantara Atlas services

We revised wording in the manuscript to state that products are made available via Nusantara Atlas, avoiding any implication of manual upload steps.

“The authors rely on several independent systems (Google Earth Engine, FIRMS, and a web visualization platform), yet the manuscript does not explain how components are programmatically integrated”

Supplementary Methods S1 now describes the integration pipeline, including:

Python-based scheduling and orchestration

Automated data exchange between GEE, cloud storage, and post-processing services

Programmatic linkage to Nusantara Atlas

We also clarify that Google Earth Engine serves as the core execution environment, with external scripts coordinating post-processing and dissemination.

“Explicitly describe the nature of the processing system

Response:

We have followed this recommendation by:

Separating analytical methodology (main manuscript)

Operational automation & architecture (Supplementary Methods)

Providing Figure S1 (workflow/system diagram)

We also revised terminology throughout the manuscript for clarity and accuracy.

---

## [Editor Report · Decision Letter 2]

9 Mar 2026

Nationwide Monthly Burned Area monitoring in Indonesia using Sentinel-2

PONE-D-25-45724R2

Dear Dr. Gaveau,

We’re pleased to inform you that your manuscript has been judged scientifically suitable for publication and will be formally accepted for publication once it meets all outstanding technical requirements.

Kind regards,

Bijeesh Kozhikkodan Veettil

Academic Editor

PLOS One
---

## [Editor Report · Acceptance letter]

PONE-D-25-45724R2

PLOS One

Dear Dr. Gaveau,

I'm pleased to inform you that your manuscript has been deemed suitable for publication in PLOS One. Congratulations! Your manuscript is now being handed over to our production team.

Kind regards,

on behalf of

Dr. Bijeesh Kozhikkodan Veettil

Academic Editor

PLOS One